# Cavitation Wear of Basalt-Based Glass Ceramic

**DOI:** 10.3390/ma12091552

**Published:** 2019-05-12

**Authors:** Marko Pavlovic, Marina Dojcinovic, Radica Prokic-Cvetkovic, Ljubisa Andric, Zoran Ceganjac, Ljiljana Trumbulovic

**Affiliations:** 1Faculty of Technology and Metallurgy, University of Belgrade, Karnegijeva 4, 11000 Belgrade, Serbia; rina@tmf.bg.ac.rs; 2Faculty of Mechanical Engineering, University of Belgrade, Kraljice Marije Street 16, 11000 Belgrade, Serbia; rprokic@mas.bg.ac.rs; 3Institute for Technology of Nuclear and Other Mineral Raw Materials, Franchet d’Esperey 86, 11000 Belgrade, Serbia, lj.andric@itnms.ac.rs; 4High Technical School of Professional Studies, 34300 Arandelovac, Serbia; zorancega@ptt.rs; 5High Business Technical School, 31000 Uzice, Serbia; ljiljanatrumbulovic@gmail.com

**Keywords:** basalt-based glass ceramics, cavitation wear, mass loss, image analysis

## Abstract

This paper examines the possibility of using basalt-based glass ceramics for construction of structural parts of equipment in metallurgy and mining. An ultrasonic vibration method with a stationary sample pursuant to the ASTM G32 standard was used to evaluate the possibility of the glass ceramic samples application in such operating conditions. As the starting material for synthesis of samples, olivine–pyroxene basalt from the locality Vrelo–Kopaonik Mountain (Serbia) was used. In order to obtain pre-determined structure and properties of basalt-based glass ceramics, raw material preparation methods through the sample crushing, grinding, and mechanical activation processes have been examined together with sample synthesis by means of melting, casting, and thermal treatment applied for the samples concerned. The mass loss of samples in function of the cavitation time was monitored. Sample surface degradation level was quantified using the image analysis. During the test, changes in sample morphology were monitored by means of the scanning electronic microscopy method. The results showed that basalt-based glass ceramics are highly resistant to cavitation wear and can be used in similar exploitation conditions as a substitute for other metal materials.

## 1. Introduction

Basalt belongs to a group of extremely hard volcanic stones. Basalt-related research from the Vrelo–Kopaonik deposit showed that it is a compact core of the basic volcanic stone with texture varying from massive to fluid. This stone rock is an easily melted material that can be used as a raw material for both glass and glass ceramics production with specific mechanical properties, high strength and low abrasiveness [1,2,3,4,5]. The basic oxides determining the quality of basalt melt in the glass ceramic products are SiO_2_, Al_2_O_3_, CaO, MgO. Basalt stones of this deposit mainly include these components so that these can be used as a raw material for basalt products. If necessary, their composition may be amended with addition of natural carbonates (dolomite, limestone), metallurgical dross, and other ingredients [6].

Favorable technical properties of basalt make it an important raw material for building applications, road constructions, and all types of traffic and fast-track railway applications [7]. As an architectural construction stone, it can be used as a cover material for external and internal horizontal and vertical surfaces, decorative furniture, dishes, decorative glazes of various ceramics, and other products [8,9,10]. Raw basalt and basalt-based glass ceramic products are used as substitutes for natural stone used for coating devices and worn parts of various plants, especially in cement industry—for mineral raw materials container storages, coke baskets and various ores, classifiers, and mixers [4,9]. Basalt is a cheap and widespread raw material used in production of glass and glass ceramics [3,4,5,11], as well as for synthesis of new materials and products such as basalt wool, basalt fibers, basalt plastics, basalt armatures, and composite materials widely used for manufacturing parts and equipment in mechanical engineering, shipbuilding, and construction industries [12,13,14,15,16]. Basalt stone processing technology is an environmentally-friendly one and the products respectively obtained are not carcinogenic [2].

Basic basalt properties, which influenced its selection for exploring the possibilities of application in engineering practice, as substitute for metallic materials are: density 2460–2960 kg/m^3^; basic amount of glass 10–15%; melting point 1300–1400 °C; high hardness 6.5–7 Mohs scale; compressive strength 80 MPa; porosity 3.78%; hygroscopicity 1–4%; moisture content 1.2%; high corrosion resistance, high resistance to frost; wear resistance; high resistance to acids, bases, and heat; ecological and hygienic quality [2,3,4,5,6,7,8]. In industrial processes, equipment parts are under the influence of high temperatures and pressures, wear, corrosion, fluid flow, reactive suspensions, and cavitation.

Cavitation is a kind of wear and represents formation, growth, and implosion (collapse) of steam or vapor gas bubbles in a flowing fluid [17,18,19,20,21,22,23,24,25]. Implosion of bubbles is caused by shock waves and micro-jet-stress concentrators which energy is either dissipated within the liquid or absorbed by a solid contact surface [26,27,28,29,30,31,32,33]. It has been shown that the impact formed by collapsing cavitation bubbles have amplitudes between the strength values on the boundary of large elongations and the tensile strength of the material, thus contributing to formation of elastic/plastic deformation and surface hardening. Impulse pressures with the amplitude larger than the tensile strength of the material cause damage and loss of the material mass, i.e., cavitation erosion. During cavitation in collapsing bubbles, high temperatures and pressures are locally produced (approximately 5000 °C and 1000 bar) in a very short time sequence (less than 1 μs) [17,24,33].

In order to evaluate the possibility of using basalt for the production of parts of industrial equipment, the cavitation resistance of raw basalt samples and basalt-based glass ceramic was determined in this paper. An ultrasonic vibration method (with a stationary sample) was applied according to the ASTM G32 standard [34]. The change in mass of samples was observed as a function of the cavitation time. In addition, the change in structural parameters and the resistance of samples determined by means of the image analysis program—Image Pro Plus—were monitored on photographs of the sample surface taken during exposure of the same to the cavitation effect [35]. On the basis of the results obtained, a comparison of properties among the samples tested was conducted to enable assessment of their prospective applications in the given exploitation conditions.

## 2. Materials and Methods

For examination of cavitation resistance, raw basalt (RB) and basalt-based glass ceramic samples (GCB) were used. Raw basalt samples, sized at (15 × 15 × 15) mm were cut from the selected basalt stones from Vrelo–Kopaonik deposit. Table 1 shows chemical composition of raw and basalt-based glass ceramic samples.

Basalt-based glass ceramic was obtained by raw basalt stones having been melted at a temperature of 1250 °C and test plates sized at (200 × 150 × 15) mm having been casted in a sand mold. In order to reduce internal stresses of the casted test plates, these were heated up at a temperature of 850 °C/2 h and then gradually cooled down to room temperature of the oven.

X-ray diffraction analysis was performed by the X-ray diffractometer, model PW-1710 (Philips Analytical, Almelo, The Netherlands). It involved a curved graphite mono-chromate meter and a scintillation counter. Intensities of the diffracted CuKα of the X-ray radiation (λ = 0.154178 nm) were measured at room temperature with intervals of 0.02° (2θ), over the time of 1 s, within the range from 4° to 65° (2θ). The X-ray tube was under the voltage of 40 kV and current of 30 mA, while primary and diffracted rays’ slots were 1° and 0.1 mm.

Morphology of damaged surfaces of the sample was analyzed by means of a scanning electronic microscope Joel JSM 6610 LV (JEOL, Tokyo, Japan). In order to improve conductivity, the sample was vapoured by gold powder.

An ultrasonic vibration method (with stationary samples) was used to test cavitation resistance according to both the ASTM G32 standard [34] and the procedure described in earlier works [24,31,32]. The ultrasonic vibration method with a stationary sample for cavitation is used when working with brittle specimens which cannot be threaded. In this case, the sample was fixed during the cavitation test with holders located at the bottom of the water bath. According with ASTM G32, three test specimens of basalt were used for the cavitation test and an average value of the measurements was taken as results for diagram which shows the relation between the mass loss of samples and testing time. According to ASTM G32, cavitation rate of the test material was calculated as the total mass loss of the sample after the total testing time. During the test, mechanical vibration concentrator was immersed in a water bath at a temperature of 25 ± 1 °C. The sample tested was placed underneath the front surface of vibration concentrator with a 0.5 mm gap. Frequency of mechanical vibrations was 20 kHz, the amplitude was 50 µm. A strong cavitation zone was formed below the concentrator front surface and the stationary-tested sample. The water from the water bath was cooling the sample to keep its temperature at the same level. A constant water flow created a pressure field stimulating implosion of cavitation bubbles on the test sample surface. The selected sample time (min) was: 15, 30, 60, 120. After each test sequence, mass loss of the sample was measured using analytic accuracy balance of 0.1 mg.

Taking into account that structure and properties of basalt samples do affect cavitation resistance, changes in structure of the sample surface were analyzed during the test. Photos of samples were taken before, during and at the end of the test. Image Pro Plus 6.0 software package tools (Media Cybernetics, Rockville, MD, USA) were used for image data processing and quantification of the morphological characteristics of selected objects. Results of the analysis enabled examination of mechanisms of sample surface destruction whereas such examination referred to monitoring the following damage indicators during the test sequence for the cavitation process: level of degradation of the sample surface, (P/P_0_, %, where P_0_ refers to the reference surface with no damage occurrence while the P-value represents a sample surface damage done during the test), number of individual pits formed, N_p_, medium area of pits formation, P_av_ (mm^2^), new pits formation, pit growth, and/or interconnection. Image samples were analyzed using red, blue, and green channels. The best resolution between damaged and undamaged surface was achieved using red color channels. Assessment of the tested samples behavior in reference to cavitation effect was analyzed on the basis of correlation between the test results and the basalt structure and properties. All results of damage to surface of the samples during cavitation activity are illustrated by respective diagrams.

## 3. Results and Discussion

### 3.1. Phase Composition of Basalts Samples

Mineral composition of raw basalt (RB) and basalt-based glass ceramics (GCB) is as follows: plagioclases, pyroxenes, olivines, Figure 1. The most prevalent minerals in the RB sample are basic plagioclases, while pyroxene (augite) and olivine are less present, Figure 1a. In the glass-ceramic sample (GCB), the most common are pyroxene, and less plagioclases and olivine, Figure 1b.

### 3.2. Micro-Structural Analysis of Basalt Samples

In Figure 2, Scanning electron microscopy (SEM) micro-photos of both RB and GCB samples are shown as before the cavitation process. Raw basalt, as the base of the sample tested, Figure 2a, was formed from a microcrystalline plagioclase with microlitic structure. Fenocrystals, olivines, rhombic pyroxene, and rarely basic plagioclases have been identified. The sample is limonitis, partially. Basalt stone structure is represented with olivine–pyroxene basalt. The basis of the tested GCB sample was cryptocrystalline with presence of small tiny crystalline crystals, Figure 2b. Sample structure is a non-homogenous one and consists of different aggregates with a clear boundary in between. During the cryptocrystalline glass base of basalt, some crystals are thermally altered.

Figure 3 shows the structure of RB and GCB sample as recorded on pole-selecting microscope. Large olivine crystals are incorporated into the base mass of plagioclases, Figure 3a. Phenocrystals of basic plagioclases are elongated, Figure 3b. The structure of GCB sample is a cryptocrystalline glass base, Figure 3c, in which the transformed spinels and pyroxenes including bubbles are partially filled with glass, Figure 3d.

The structure of RB and GCB samples contains bubbles of various sizes filled with air or glass, Figure 4. Bubbles present on the sample surface cause surface roughness and appearance of pits. RB sample contains a large number of tiny bubbles, Figure 4a, while the GCB samples contained bubbles of larger dimensions that are embedded in the cryptocrystalline–glass base, Figure 4b.

During cavitation test, changes of bubbles contained in the basalt base and the presence of pits on both RB and GCB samples surface were monitored.

### 3.3. Implementation of Image Analysis on Comparison of Cavitation Erosion Degradation of Basalt Samples

Figure 5 shows the results of examining samples of raw basalt and basalt-based glass ceramics during cavitation erosion test. For the analysis of the cavitation wear of the samples, the change in the sample mass in the function of the exposure time and the changes occurring on the surface of the samples (the level of surface degradation, the number of formed pits, and the mean surface of the formed pits) was observed.

### 3.4. Mass Change

Mass loss measurement applied for samples subject to cavitation during the test sequence is shown in Figure 5a. Mass loss resulting from cavitation damage is displayed on the y-axis, and time intervals are displayed on x-axis. It has been shown that GCB samples have a significantly higher cavitation erosion resistance with an average cavitation rate of 0.03 mg/min when compared to RB samples with a cavitation rate of 0.74 mg/min.

Having analyzed erosion progression of GCB samples, it may be concluded that the mass loss is low; within the first 15 min, mass loss was 1.29 mg and then slightly increased to 3.53 mg during 120 min of exposure. In RB samples, it is evident that for the first 15 min, mass loss of RB sample was up to 15 mg; as the exposure time increases, cumulative mass loss of the sample gradually increases in an almost linear manner up to 88.5 mg during 120 min of exposure.

Higher erosion rate in RB samples in comparison to GCB samples can be explained by means of a rough structure of the olivine-pyroxene basalt of RB samples and through their comparison with compact structure of GCB samples which contributes to an increased resistance of GCB samples to the cavitation effect.

### 3.5. Image Analysis

Photos of both RB and GCB samples were taken before and during the cavitation erosion test, as shown in Figure 6. It was noted that GCB sample presented lower scope of surface damage than RB samples. There were almost no dimensional changes in pits that were present on the sample surface prior to test. Profile lines of GCB sample are uniform with individual peaks present at the same locations on the surface of the sample. Presence of individual pits caused by presence of bubbles in the structure were identified prior to the test.

On the RB sample, initial pits on the surface and the roughness present (during *t* = 0 min) were changing and increasing in size during cavitation exposure time, as it was reflected on RB sample profile lines producing major changes of the same during the test. These profile lines indicate that degradation was happening in the center of the sample surface, as the intensities of the profile lines edges were changing and increasing between a 15 min exposure and a 120 min one where significant surface area damage of the RB sample arises. The results shown in Figure 6 are in line with the results of surface damage of RB and GCB samples as determined with application of image analysis on photos of sample surfaces taken during the cavitation time. The same were then processed and analyzed using the Image Pro Plus software, as shown in Figure 5b.

At the end of 120 min test of GCB samples, minor changes were observed on the surface of the sample, with a significantly lower number of small pits in reference to RB samples where the surface was damaged to a higher extent with presence of a number of pits here and there joined to create larger and deeper pits. This is in line with the results of a gradual sample mass loss during the test, Figure 5a. At the end of cavitation exposure, GCB sample surface damage amounts to 12%, while RB sample damage is over 35.9%, as shown in Figure 5b. This is in line with the results obtained for the average surface of pit formation, P_av_ shown in Figure 5c.

Figure 5d shows the dynamics of pit formation on the RB and GCB samples surface during exposure to cavitation effect.

In the RB sample structure there are numerous minor bubbles (Figure 4a) forming multitude of small pits on the sample surface thus causing an increased surface roughness (Figure 6, RB sample for *t* = 0 min). With the activity of cavitation process, number of newly formed pits is gradually increased in up to 60 min of exposure; then, after 120 min, number of pits is slightly lowered to indicate that the pits get connected in some places, Figure 5d. Increment of the number and size of pits—which in some places get connected to form larger and deeper pits—causes an increased surface damage of the samples, Figure 5c. Larger and deeper pits on the damaged RB sample surface allow for the focus of energy of the shock waves caused by implantation of cavitation bubbles intensifying the cavitation effect. An analysis of the RB samples image showed that generation of a larger number of smaller pits on the sample surface and their merging into larger and deeper pits contributed to the extent of damage of the sample surface. After 120 min of cavitation activity, damage of surface amounted to 35.9%, Figure 5b.

In the GCB sample structure, there are individual larger bubbles (Figure 4b), as well as individual larger or smaller pits before the beginning of cavitation activity, (Figure 6, GCB, for time *t* = 0 min). On the surface of the GCB sample cavitation changes are noticed after 30 or 60 min of cavitation activity. From the beginning of cavitation activity up to 30 min, a very small number of tiny new pits are formed; up to 60 min the number of pits is rapidly decreasing, most likely due to their interconnection; from 60 to 120 min of exposure the number of formed pits is slightly increased, Figure 5d. The middle surface of formed pits is gradually widened, Figure 5c. Analysis of GCB samples showed that initial pits on the sample surface were most likely due to presence of bubbles in the structure; further, these did not change during exposure, as can be seen on photos of GCB samples during the test; a smaller number of pits caused less damage to the GCB sample surface. Therefore, surface damage is below 12% after 120 min of cavitation activity, Figure 5b.

The mechanism of pit formation and growth on the RB samples surface could be monitored by having the image analyzed, starting from the early phase of the cavitation process then after 15 min of exposure to the end of the test in duration of 120 min. In the GCB samples, tiny pits formed on sample surface after 30 min of cavitation activity changed very little in shape and size by the end of the test. Changes in morphology of both RB and GCB sample surfaces over the test time were monitored by the scanning electronic microscopy method, Figure 7 and Figure 8.

Figure 7 shows a change in the surface area of the RB samples during exposure to the cavitation process. By performing a 15 min cavitation process, the existing roughness of the surface increases and the shallow pits form (Figure 7a). The pit in the olivine-pyroxene basalt structure is sharp edged, most likely due to erosion of edges of the crystal around the existing bubble (Figure 7b). A 30 min exposure influenced increment of the sizes of the pits formed (Figure 7c). Presence of pits with surfaces eroded due to removal of the grain and mass loss caused by cavitation effect is shown in Figure 7d. Cavitation activity lasting for 60 min influenced further degradation of sample surface. A number of new pits were formed while the existing pits increased having eroded their surface to lead to a further mass loss of the samples concerned (Figure 7e,f). At the end of the test—with cavitation effect for 120 min—the RB sample surface was highly deformed (Figure 7g). The crystals of olivine and pyroxene minerals, respectively present, eroded in different ways (Figure 7h).

On the Figure 8, at the beginning of the test (in 15 min), there was no change in the sample surface. The pits existing on the surface were caused by bubbles in the crystalline base of basalt (Figure 8a,b). With the cavitation effect, during 30 min, there was no change in the sample surface. The bubbles existing in the cryptocrystalline structure of GCB samples were filled with glass mass (Figure 8c,d). Cavitation activity during 60 min shows erosion surface with pits occurred near the bubble, Figure 8e and surface deformation of the pits filled with glass mass, Figure 8f. At the end of the 120-min test, erosion of pits occurred near the bubble in the base of basalt, Figure 8g,h.

## 4. Conclusions

The effects of the ultrasonic vibration method for determination of cavitation damage of basalt were studied. The aim was to determine the possibility to use basalt-based glass ceramics to obtain certain structural elements of the metallurgy and mining equipment exposed to high cavitation loads. Several approaches have been used to monitor the level of damage to the samples: change in mass loss, measure the level of surface degradation, form pits, and change the morphology of the damaged surface, line profiles. Analysis of mass loss and progression of erosion on the sample surface during the cavitation process showed different effects of cavitation damage of raw basalt samples and samples of basalt-based glass ceramics obtained by melting and casting of basalt stones.

The cavitation erosion of basalt-based glass ceramics samples is characterized by the appearance of small pits located near the bubbles in the basalt structure and their number increases at low speed during exposure. It has been shown that glass ceramic samples based on olivine–pyroxene basalt from the test deposit can be applied in conditions in which high cavitation loads are expected.

Raw basalt samples show much loss resistance to cavitation wear. The mechanism of damage is characterized by the appearance of number of pits near the bubbles in the basalt structure and their number grows at a higher speed along with a greater mass loss during exposure to cavitation. The selected image analysis method and the standard mass loss measurement produce are reliable and fully contribute to the characterization of basalt samples during cavitation wear and can be used for quick and reliable selection of materials for use in these conditions.

## Figures and Tables

**Figure 1 materials-12-01552-f001:**
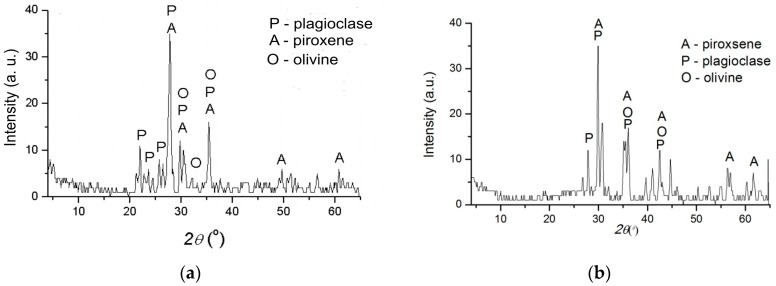
X-ray diffraction (XRD) of basalt samples: (**a**) raw basalt; (**b**) basalt-based glass ceramics.

**Figure 2 materials-12-01552-f002:**
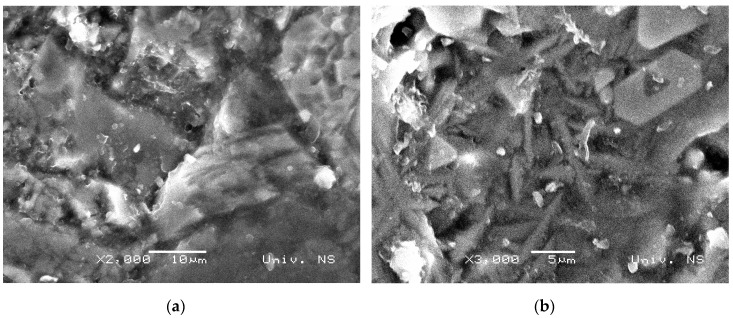
Scanning electron microscopy (SEM) microphotographs of basalt samples: (**a**) raw basalt; (**b**) glass-ceramic based on basalt.

**Figure 3 materials-12-01552-f003:**
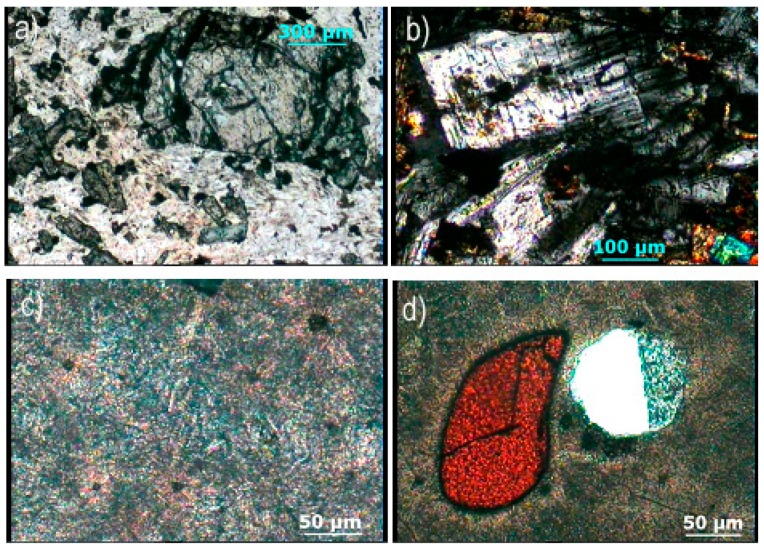
Microphotographs of raw basalt (RB) and basalt-based glass ceramic (GCB) samples: (**a**) crystals of olivine in the basis of plagioclases; (**b**) elongated phenocrystals of plagioclases; (**c**) cryptocrystalline glass basis of GCB sample; (**d**) altered spinel with bubble partially filled with glass.

**Figure 4 materials-12-01552-f004:**
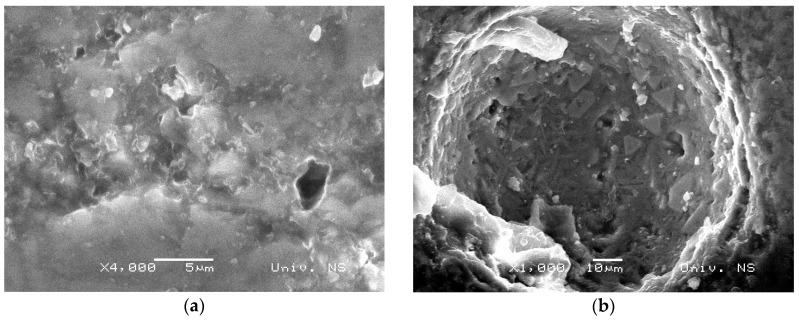
The present bubbles in the basalt structure: (**a**) raw basalt (RB) sample; (**b**) basalt-based glass ceramic (GCB) sample.

**Figure 5 materials-12-01552-f005:**
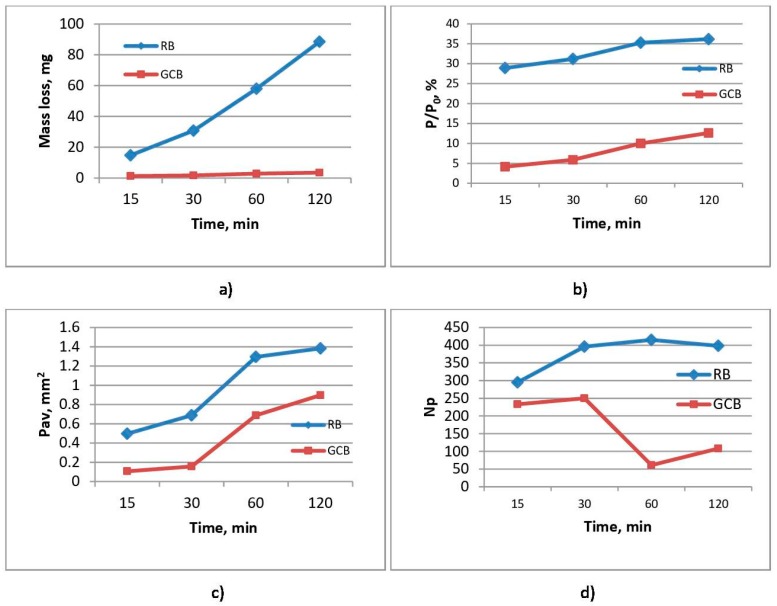
Results of raw basalt (RB) and basalt-based glass ceramic (GCB) samples during the cavitation erosion testing: (**a**) mass loss of samples; (**b**) surface degradation level; (**c**) average area of the formed pits; (**d**) number of the formed pits.

**Figure 6 materials-12-01552-f006:**
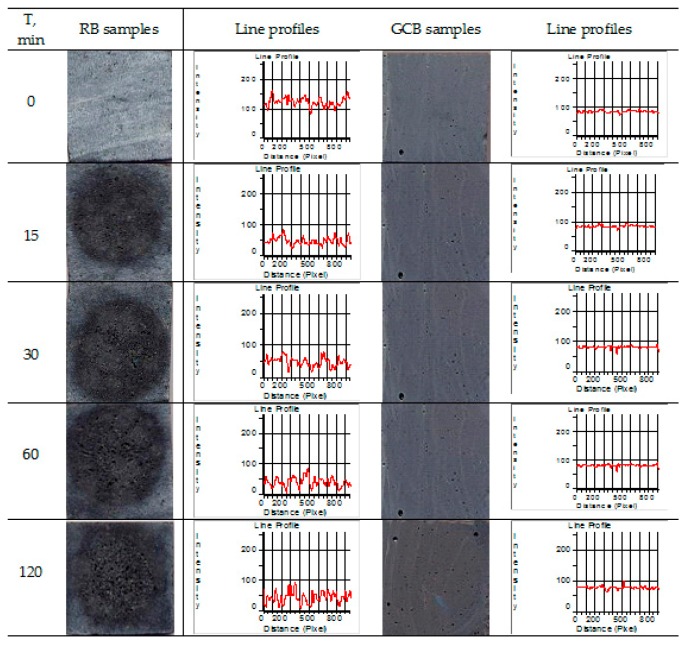
Photos of both raw and glass ceramics based on basalt samples before and during cavitation erosion test after implementation of red filter and respective line profiles.

**Figure 7 materials-12-01552-f007:**
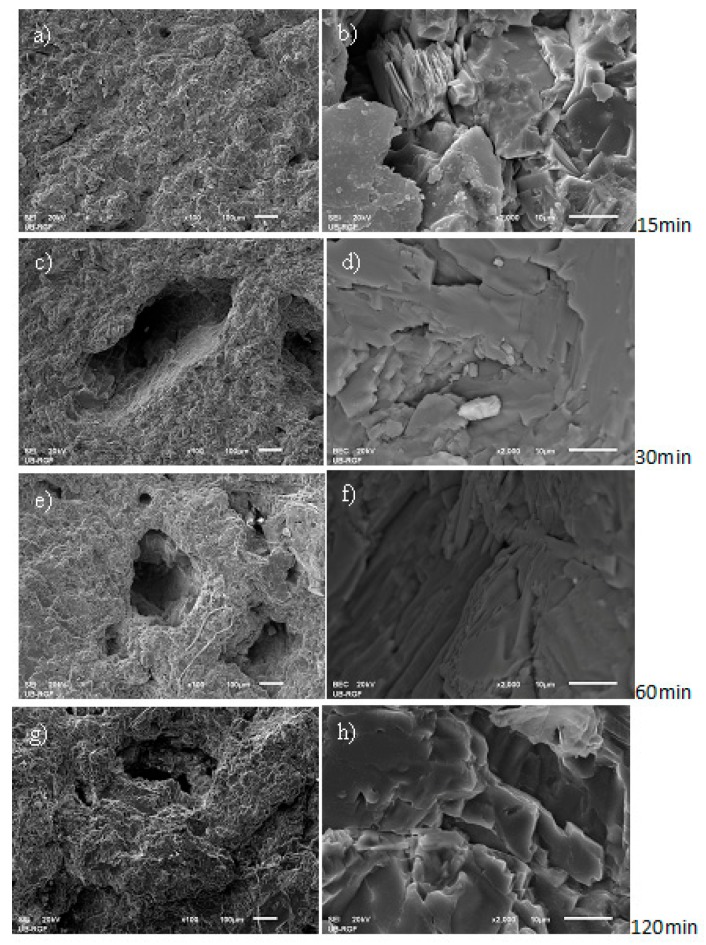
Scanning electron microscopy (SEM) micro-photos of deformed raw basalt (RB) sample surfaces with different magnitudes (500× left, 1000× right) and cavitation effects times: (**a**,**b**) 15 min; (**c**,**d**) 30 min; (**e**,**f**) 60 min; (**g**,**h**) 120 min.

**Figure 8 materials-12-01552-f008:**
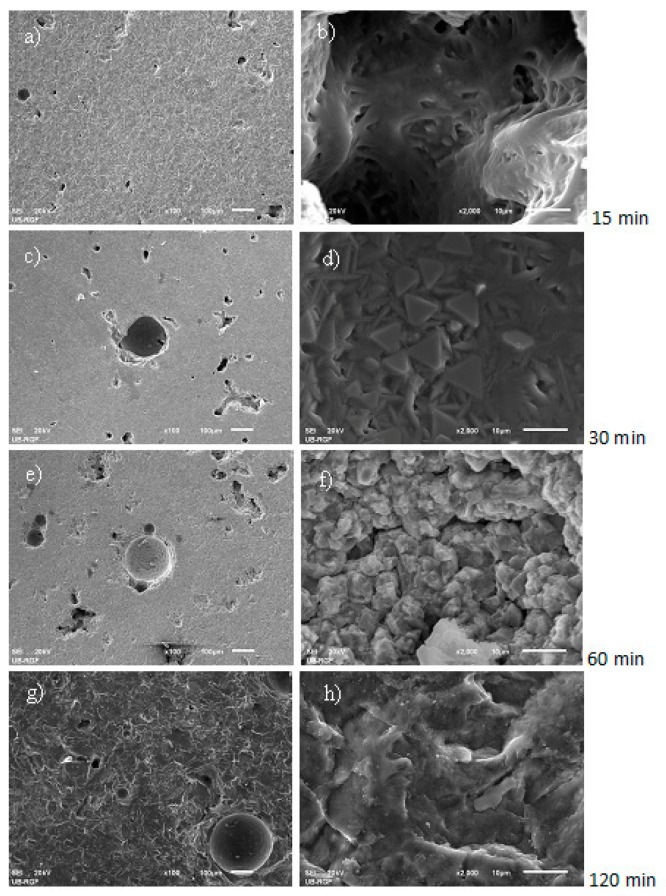
Scanning electron microscopy (SEM) micro-photos of deformed basalt-based glass ceramic (GCB) sample surface with different magnitude (500× left, 1000× right) and cavitation effect times: (**a**,**b**) 15 min; (**c**,**d**) 30 min; (**e**,**f**) 60 min; (**g**,**h**) 120 min.

**Table 1 materials-12-01552-t001:** Chemical composition of raw basalt and basalt-based glass ceramic samples, (wt.%**).**

Sample	SiO_2_	Al_2_O_3_	Fe_2_O_3_	FeO	MgO	CaO	Na_2_O	K_2_O	TiO_2_
Raw basalt	56.21	18.61	1.15	2.97	3.40	7.78	4.73	3.37	1.11
Basalt-based glass ceramic	52.78	16.97	1.87	7.52	7.56	8.94	1.79	0.89	1.26

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
