# Peer review of "Cavitation Wear of Basalt-Based Glass Ceramic"

_materials, 2019, doi:10.3390/ma12091552_

Reviewer 1 Report

Attached.

Author Response

We send you Word PDF file

Reviewer 2 Report

The manuscript reports on the comparison of the behaviour of basalt and cast glass-ceramic (made from basalt) pieces versus ultrasonic cavitation in 25°C water for 15 to 120 min. Raw materials are characterized by XRD and SEM. Morphology of damaged surfaces was also studied by SEM and the degradation compared by mass loss, roughness profile (pits formation, pits number).

The work is original and deserves publication after the following weaknesses have been addressed.

A more concise redaction is needed. The number of micrographs (Figs 10 and 11) must be reduced to avoid redundancy and their quality/contrast improved. Composition of the glass ceramic samples must be also given.

The authors don’t try to explain the different behaviours observed for raw basalt and for the glass ceramic. Water, especially in cavitation is very corrosive and may induce insertion of protons by replacement of Na+ and K+ ions (see e.g. Solid State Ionics 70/71, 1994, 109; Solid State Ionics 179, 2008, 2142; Ceramics Int. 41, 2015, 14137 and references herein) that is very detrimental for the mechanical properties (formation of cracks, crumbling). A comparison of the XRD patterns of pristine materials and of materials exposed to cavitation after a thermal treatment at about 700°C will allow a better characterization of the degradation induced by cavitation. Thermogravimetry will be also very useful.

For a better comparison merge Figs 5, 7, 8 and 9 in a single figure.

Author Response

We send you Word PDF file

Reviewer 3 Report

The article "Cavitation wear of basalt-based glass ceramic by M. Pavlovic, M. Dojcinovic, R. Prokic-Cvetkovic, L. Andric, Z. Ceganjac, L. Trumbulovic describes the possibility to use basalt-based glass ceramics for construction of structural parts of equipment in metallurgy and mining.  The authors compare the cavitation resistance of natural basalt from the Vrelo-Kopaonik deposit and the same basalt after melting, casting and heat treatment (basalt-based glass ceramic). The authors in their conclusions state that the cavitation resistance of basalt-based glass ceramic is much better than that of natural basalt. The work presents a scientific value. In my opinion the work is suitable for publication after making corrections.

My comments:

1.       In line 110, the authors write: "In this way, the test was not exposed to mechanical strains during the test". However, they do not describe how the sample was fixed during the cavitation test, so you can not know whether this sentence is true.

2.       In line 165-167 authors write: "It has been shown CGB that samples have a significantly higher cavitation resistance with cavitation rate of 0.0294 mg / min when it compared with RB samples and cavitation rate of 0.738 mg / min”. In support of this statement, the authors cite Figure 5. However, Figure 5 does not present changes in the rate of cavitation erosion but changes in cumulative mass loss during the cavitation test. In addition, it is not known what rate of cavitation erosion the authors write: is the maximum rate, or maybe the average rate? In this sentence, the statement "cavitation resistance" is also incorrect. The authors should write "cavitation erosion resistance".

3.       In lines 172-173, the authors write: "In RB samples, it is evident that incubation period is short at an early stage of damage ...". Figure 5 does not show the incubation period. The authors of the first measurement of the mass loss made after 15 minutes of cavitation loads. After this time, measurable losses in mass already occurred. If the authors wanted to determine the duration of the incubation period, they should take measurements of the loss of mass, for example every 2-3 minutes.

4.       In line 179 The authors write "... CGB samples with a very high hardness ..." The authors should measure and state what hardness RB and CGB samples had.

5.       Comment to figure 6. The authors should indicate the length of the measurement line - information that the length is 1000 pixels is not very readable. The authors also do not inform how they selected the measurement line, whether it was the geometrical center of the surface sample or other place.

6.       In the conclusions, the author writes (line 272): "the incubation period in the early phase of cavitation damage is short ...". Commentary - see point 3

7.       In the conclusions, the author writes (line 284): "…with very high hardness”. Commentary - see point 4.

8.       In the conclusions, the author writes (line 295): "…cooling processes that eliminate internal stresses and reduce brittelness of samples”. Commentary - This is an unauthorized statement because the authors did not investigate either brittleness or own stresses

Author Response

We send you Word PDF file

Round  2

Reviewer 2 Report

Layout has been improved and some issues addressed. However its a pitty that the authors don't try to analyze the corrosion mechanism by water taking into accountappropriate literature

Author Response

The aim of the research in this paper was to show the possibility of using basalt and glass-ceramics based on basalt in cavitation conditions. The corrosion resistance of these materials was one of the essential criteria for their selection in this test. This was the reason why cavitation corrosion was not included in laboratory test, but the behavior of materials from the aspect of damage was monitored. The suggestion of the reviewer is significant, but in this case the analysis of the corrosion mechanism in application of basalt and the glass-ceramics based on basalt in cavitation conditions require the setting up of a new experiment.
